# On Directional Dendritic Growth and Primary Spacing—A Review

**Joel Strickland, Bogdan Nenchev and Hongbiao Dong *** 

School of Engineering, University of Leicester, Leicester LE1 7RH, UK; jcjs2@leicester.ac.uk (J.S.);
bn55@leicester.ac.uk (B.N.)

*** Correspondence: h.dong@leicester.ac.uk; Tel.: +44-(0)-116-252-2528

**Abstract:** The primary spacing is intrinsically linked with the mechanical behavior of directionally solidified materials. Because of this relationship, a significant amount of solidification work is reported in the literature, which relates the primary spacing to the process variables. This review provides a comprehensive chronological narrative on the development of the directional dendritic growth problem over the past 85 years. A key focus within this review is detailing the relationship between key solidification parameters, the operating point of the dendrite tip, and the primary spacing. This review critiques the current state of directional dendritic growth and primary spacing modelling, briefly discusses dendritic growth computational and experimental research, and suggests areas for future investigation.

**Keywords:** directional solidification; microstructure; dendritic growth; primary spacing; modelling

## 1. Introduction

Dendrite is a descriptive word derived from the Greek, "dendron", which means tree. In metallurgy, dendrites are arborescent crystalline structures that grow by diffusion-limited heat and mass transfer. At their origin, is undercooling below the freezing point of the solid and nucleation of a crystal. The accumulation of solute ahead of a growing interface can cause constitutionally undercooled zones [1,2], which under appropriate circumstances can become unstable and the solidification morphology dendritic. In metals, the surface energy is typically anisotropic and certain growth directions are energetically favoured. In most cubic metals, the preferential growth directions are <100>, which results in crystal growth parallel and opposite to the heat flow. As a result, metallic cubic dendrites exhibit crystallographically-related features, such as primary trunks, secondary and tertiary side arms, and sometimes arms of even higher order.

In directional solidification, the heat transfer is constrained through the solid, which results in primary trunk growth aligned opposite to the direction of heat flow. In the regime where the solid-liquid interface is strongly morphologically unstable, arrays of dendrites evolve from homogenous starting compositions, into complex spatio-temporal patterns far from equilibrium [3]. In the literature, single crystal patterns are classified as either square, hexagonal, or random [4]. The variation in composition between the advancing dendrite and the surrounding interdendritic region gives rise to microsegregation within the solidified crystal [5,6]. Normal to the dendrite growth direction, this segregation is characterised by the primary spacing, $\lambda_1$. The primary spacing controls the maximum length scale for the microsegregation [7], the solutioning heat treatment times [8,9], and the mechanical properties of the directionally solidified material [10–24]. In addition, the $\lambda_1$ directly influences the mushy zone convection, the formation of low melting point secondary phase eutectics, as well as incoherent precipitates and pores in the interdendritic region [25–32]. Consequently, the mechanical

properties of unidirectional solidified crystals are strongly dependent on the temperature and convection within the melt, as this controls the concentration of solute at the solid–liquid interface.

In the literature, a significant amount of steady state solidification work is reported that characterises the relationship between primary spacing, alloy composition, $C_0$, thermal gradient, $G$, and tip growth velocity, $V$ [33–41]. In directional solidification, the $G$ and $V$ can be independently controlled, so one may study the dependence of $\lambda_1$, on either $G$ (at constant $V$) or $V$ (at constant $G$). Over the years, the fundamental understanding of the relationship between dendrite tip growth and primary spacing has improved considerably. Theoretical models and extensive experimental studies have now established criterion to determine dendrite tip radius as a function of the growth parameters [42–44]. During the same period, primary spacing modelling has undergone its own arduous empirical and theoretical journey, whereby researchers have concentrated on relating the tip growth kinetics to the resultant microstructural patterns [40,41,45–50]. As a result, the primary spacing was linked to the process variables by the following non-linear steady state relationship:

$$\lambda_1 = AC_0^{0.25}V^{-0.25}G^{-0.5}. \tag{1}$$

Current day solidification science has developed into a massive international community. The field has diverged into a variety of casting, characterisation, modelling, and defect prediction-specific areas; for a broader overview, the reader is directed to references [51,52]. Directional dendritic growth and primary spacing modelling still piques the interest of a significant number of researchers, with many tantalising challenges and tangible rewards for society still available. For this reason, the aim of this review is to provide a comprehensive chronological narrative on the development of the directional dendritic growth problem over the last 85 years. The authors elucidate the relationships between key processing parameters, the operating point of the dendrite tip, and the primary spacing. This review critiques the current state of primary spacing modelling, briefly discusses dendritic growth computational and experimental research, and concludes by suggesting directions for future research.

## 2. The Operating Point of a Dendrite Tip (1935–1999)

Modelling of directionally solidified microstructural patterns, first requires a theoretical understanding of the growth of an isolated dendritic tip and determination of a steady state solid–liquid interface shape (Figure 1). In 1935, Papapetrou [53] proposed that a parabolic interface satisfied the shape preserving condition. The author reasoned that if only part of the latent heat was rejected ahead of the growing dendrite, a fixed concentration profile and shape would be maintained.

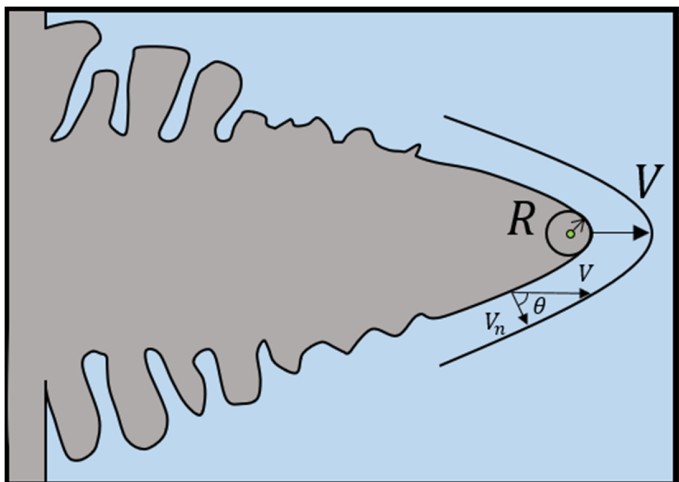

**Figure 1.** Shape preserving condition for steady state growth of a dendrite tip. $V$ is the dendrite tip growth velocity; $R$ is the dendrite tip radius; $V_n$ is the normal velocity required to maintain the steady state shape.

In 1947, Ivantsov [42] provided elementary mathematical treatments for an isothermal parabolic plate-like dendrite (2D) and a paraboloid of revolution (3D needle crystal) with an isoconcentrate interface. Ivantsov solved the self-consistent shape preserving condition for steady state dendritic tip growth with the following equations:

for the 2D case,

$$\Omega_C = \sqrt{\pi P_c}\ exp^{P_c}\left(1 - erf\ \sqrt{P_c}\right), \tag{2}$$

and the 3D case,

$$\Omega_C = P_c exp^{P_c} E_1(P_c), \tag{3}$$

where, $P_c$ is the solute Péclet number ($VR/2D$); $V$ is the dendrite tip growth velocity; $R$ is the dendrite tip radius; $D$ is the liquid solute diffusivity; $E_1(P_c)$ is the exponential integral; $\Omega_C$ is the dimensionless solute undercooling $(C_t - C_0/C_t(1-k))$; $C_t$ is the composition in the liquid at the dendrite tip; $C_0$ is equilibrium alloy composition; $k$ is the solute partition coefficient. Later, the Ivantsov analysis was extended by Horvay and Cahn [54] to paraboloids with elliptical cross-sections. These researchers generalised the Ivantsov solutions for a paraboloid of revolution by expressing the dendritic surface in dimensionless variables and varying the eccentricity of the elliptical cross-section. In the isotropic case, these quasi-stationary Ivantsov solutions describe the dendritic tip shape as a smooth sphere. However, in the anisotropic case, the sphere is deformed in the direction of the anisotropy strength. For a given tip undercooling, $\Delta T$, the Ivantsov solutions produced an infinite set of $V$ and $R$ combinations that satisfied the requirement of $V \times R = constant$ (Figure 2). Therefore, for a given set of growth conditions, no unique solution was determined for the dendritic tip operating point. Thus, an additional selection constraint independent of energy transport was required.

Around the same time as Ivantsov was publishing his seminal work, Zener [55] was introducing the concepts of diffusion and capillarity in the form of equations for the lengthening of needle crystals in lamellar phase transformations. Although Zener was investigating solid-state transformations, the associated physical phenomena were in many ways like that which occurs in dendritic growth. Zener realised that the inclusion of capillarity produced a maximum in the $V - R$ curve and suggested a phase change occurred at the greatest growth rate ('Extremum Condition'), rather than at the lowest free energy. Hillert [56] found Zener's diffusion-controlled model to be inaccurate for high supersaturations and modified it by deriving an approximate analytical solution for the diffusion equation that related time and structure. The Zener–Hillert model established the theoretical foundation for diffusion-controlled growth kinetics of lamellar phases and explained the relationship between lamellar spacing, $\Delta T$, and $V$ during the phase transition.

### 2.1. The Extremum Condition

In the 1960's, Temkin [43] and Bolling and Tiller [57] pointed out the nonisothermal nature of the interface due to interface curvature and interface kinetics, which made the actual steady-state dendritic tip shape deviate from a paraboloid of revolution. The researchers subsequently included the stabilising effect of the interface energy and/or interface attachment kinetics in an attempt to provide a unique solution to the Ivantsov equations. However, the addition of capillarity did not remove the multiplicity of predicted $V$ and $R$ combinations; unfortunately, only the very small dendrite tip radii solutions were unstable (Temkin Curve—Figure 2). To overcome this problem and provide an exact solution to the Ivantsov equations i.e., determine the operating point of a steadily growing dendrite tip, many authors [43,57–61] applied the 'extremum condition' as suggested by Zener.

In 1965, Jackson and Hunt discovered low melting point organic transparent materials that froze like metals, and performed the first in-situ observations of dendritic pattern formation [62,63]. Following their pioneering work, Glicksman et al. [64,65] demonstrated that succinonitrile could realistically simulate metal solidification and investigated the current steady state dendritic tip growth theories. Their experiments confirmed that application of the extremum condition predicted $R$ over an order of magnitude larger than observed experimentally [64]; thus, a new tip selection theory

was required. Glickman's seminal work, was fundamental in validating the theory of constitutional undercooling suggested by Ivantsov [1] and Tiller et al. [2] over a decade earlier.

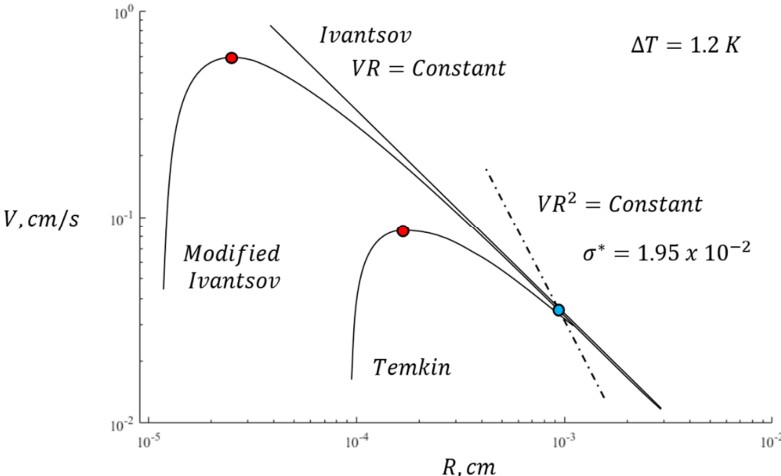

**Figure 2.** Dendrite tip growth rate, *V*, as a function of the tip radius, *R*, for constant undercooling, $\Delta T$, in pure succinonitrile. Extremum condition indicated by the red circles on the Modified Ivantsov and Temkin curves. Marginally Stable point indicated by blue circle. Plot adapted from [66].

*2.2. The Marginal Stability Condition*

In 1973, Oldfield performed the first computer calculations of dendritic behaviour [67]. Oldfield's work was strictly numerical and suggested that the dendritic tip operated between a balance of the stabilising effect of capillarity and the destabilising effect of diffusion or thermal conduction. Under the assumption of steady state heat flow, Oldfield's stability criterion is expressed as:

$$\sigma^* = \frac{2\alpha d_0}{VR^2} = 0.02, \tag{4}$$

where, $\sigma^*$ is the stability criterion; $\alpha$ is the thermal diffusivity; $d_0$ is the capillary length $\left(T_m \gamma C_p / L^2\right)$; $T_m$ is the melting point; $\gamma$ is the surface tension of the solid–liquid interface; $C_p$ is the heat capacity of the liquid phase; $L$ is the heat of fusion. It is worth noting that the Ivantsov thermal diffusion models (Equations (2) and (3)) were derived using an isothermal interface, $\gamma = 0$.

In 1978, Langer and Müller-Krumbhaar (LM–K) [68,69] applied a Mullins and Sekerka [70,71] type stability analysis to calculate the additional physical principle required to find the operating state of a steadily growing dendrite tip. According to Mullins and Sekerka, for a crystal with a planar solid/liquid interface to grow stably, the wavelength of perturbation at the interface must be smaller than:

$$\lambda_s = 2\pi \sqrt{2\alpha d_0 / V}, \tag{5}$$

where, $\lambda_s$ is the critical perturbation wavelength. LM–K suggested that dendritic tips grew at the 'marginally stable' operating point, rather than at maximum growth velocity (Extremum Condition—Figure 2). They took an Ivantsov parabola-type dendrite growing in a pure undercooled melt and considered a small departure from the parabolic tip shape caused by the interfacial energy effect [72], illustrated in Figure 3a. LM–K assumed a steady state planar interface exists at the dendrite tip and that the tip shape changes only slightly when isotropic capillarity is introduced. The criterion from equation (4) became:

$$\sigma^* = \frac{2\alpha d_0}{VR^2} = \frac{1}{4\pi^2} = 0.0253. \tag{6}$$

LM–K concluded that any tip radii smaller than the extremum point (Figure 2) would be unstable due to the increasing influence of capillarity. Their qualitative suggestion was that the dendrite tip

becomes as large as possible before becoming so large that tip-splitting occurs; it operates at the marginally stable point (Figure 2). However, the marginally stable point only confirmed that diffusion and capillarity were important characteristics of the physics behind dendrite formation. It did not provide any information regarding why a dendrite chooses the marginal stable state over other stable states [72].

In 1981, Huang and Glicksman (H–G) performed a test of Equation (6) and measured the dendrite tip $V$ and $R$ experimentally [66]. The researchers determined $\sigma^* = 0.0195$, which was within the computational uncertainty of the LM–K steady state planar tip interface model and near the lower uncertainty limit of the theory. The $\sigma^*$ predicted by Equation (6) agreed within 25% of H–G's experimentally determined values. To account for proper local curvature at the dendrite tip, H–G approximated the tip interface as a sphere (Figure 3b). They noted the wavelength of perturbation around a sphere of radius $R$ is a function of the spherical harmonic, $n$. To be consistent with the LM–K assumption that dendritic tip radius approximates to $\lambda_s$, H–G selected $n = 6$. The stability criterion in Equation (6) now reduced to:

$$\sigma^* = \frac{2\alpha d_0}{VR^2} = 0.0192. \tag{7}$$

H–G's modified spherical tip model agreed within 1.5% of their experimental results. They demonstrated that when the influence of container walls and other environmental factors were eliminated, $\sigma^*$ obtained an excellent fit with experimentally determined stability criteria.

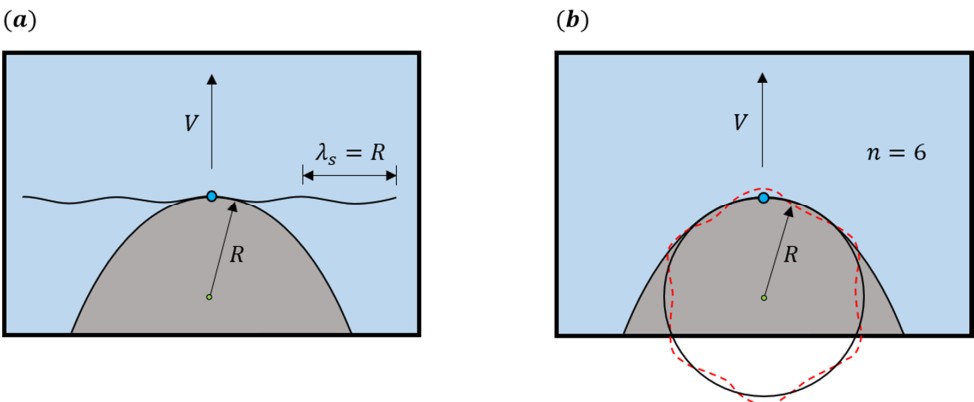

**Figure 3.** Schematics showing models for estimating dendrite tip stability: (**a**) perturbation of a planar interface at the marginally stable condition; (**b**) spherical interface model at a marginal stability condition with harmonics $n = 6$. Schematics adapted from [66].

*2.3. The Microscopic Solvability Condition*

In the 1980's, several authors [39,44,73–79] solved the self-consistent dendrite growth problem. Their equations balanced the thermal and solutal fields with the capillarity effect whilst satisfying the shape preserving condition. These researchers demonstrated that when either surface tension or surface kinetics are included, the continuous family of parabolic Ivantsov-like solutions are destroyed/eliminated. This is because the solid–liquid interface energy for crystalline materials is anisotropic, therefore, only when it was included did one of the solutions remain stable with respect to tip-splitting modes [80]. For small values of $\gamma$ the solvability condition was demonstrated to be proportional to:

$$\sigma^* = \frac{2\alpha d_0}{VR^2} \cong \sigma_0 \varepsilon^{1.75}, \tag{8}$$

where, $\sigma_0$ is a numerical constant on the order of unity, defined with help of asymptotic methods; $\varepsilon$ is a measure of fourfold anisotropy of the interface energy ($\gamma \propto [1 - \delta \cos(4\theta)]$); $\delta$ is the strength of anisotropy; $\theta$ is the angle between dendrite surface normal and its axis [51]. This new theoretical approach was designated the "microscopic solvability criterion" and provided a much firmer

fundamental basis than marginal stability (Equation (7)). The name reflected the interplay of microscopic $d_0$, which imposes a solvability condition on a family of macroscopic steady state tip solutions [81]. The theory demonstrated that marginal stability was still the main selection parameter, but the value of $\sigma^*$ depended on the crystalline surface-tension anisotropy of the material. Unfortunately, due to the difficulty in calculating anisotropy, the predicted microscopic solvability tip selection parameter was only approximate in three-dimensions. Since predictions assumed an axisymmetric dendritic tip shape, there was inconsistent agreement between theory and experimental values [81]. Although the marginal stability constant has no real physical basis, it is simple to implement and predicts the operating point of a dendrite with reasonable accuracy.

*2.4. A Test of Theory*

To examine and test the diffusion-limited dendritic growth theories, Glicksman et al. [82] proposed an isothermal dendritic growth experiment in microgravity. The purpose was to greatly reduce the influence of convective heat and mass transport on dendrite tip selection. The reduction in buoyancy-driven convection enabled Glicksman et al. to study dendrite growth via only gravitationally independent sources of heat and mass transfer. By using a low Péclet number transparent organic analogue, they were able to study 'convection free' growth and measured the true growth kinetics and morphology of dendrites. The results from various microgravity experiments [83–88] concluded that the dendrite tip can be approximated as a paraboloid of revolution, as suggested by Papapetrou [53] 60 years earlier. This fundamental work validated the pure diffusion part of the dendritic growth theory with a tip selection rule of the form of Equation (8).

## 3. Primary Spacing Selection in Directional Solidification (1979–2004)

Directionally solidified microstructures are formed of complex spatio-temporal patterns, which when viewed normal to the array growth direction, are characterised by the $\lambda_1$ (Figure 4). The range of $\lambda_1$ within a solidified component determines the distribution of inhomogeneities in the material, which significantly influences the final mechanical properties [51,52]. The $\lambda_1$ varies with alloy composition, $C_0$, and solidification process variables, such as, thermal gradient, $G$, growth velocity, $V$, and melt flow. In constrained growth, the system variables can be independently controlled, so one may study the dependence of $\lambda_1$, on either $G$ (at constant $V$) or $V$ (at constant $G$).

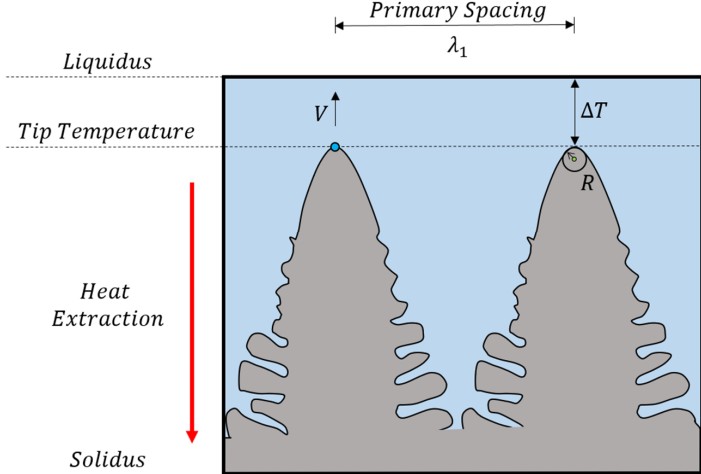

**Figure 4.** In directional solidification, most metallic cubic dendrites grow in arrays parallel and opposite to the heat flow direction. The primary spacing, $\lambda_1$, characterises twice the maximum diffusion distance between advancing neighbouring dendrites. Solute interaction between dendrites occurs within the interdendritic region and this influences the segregation pattern within a directionally solidified material. The blue dot indicates the operating point of the dendrite tip.

### 3.1. Early Primary Spacing Models

In 1979, Hunt developed the first model that linked directional solidification process variables with $\lambda_1$ [45]. The author approximated the arborescent dendritic shape as a smooth steady-state interface and derived a self-consistent interface shape using an approach developed by Brower, Brody, and Flemings [89]. In Hunt's model, the temperature and liquid composition varied only in the growth direction. However, this assumption was not valid for the derived steady state interface near the tip. To overcome this problem, the author fitted part of a sphere at the tip and obtained a relationship between $R$ and $\lambda_1$. Following this, Hunt derived a relationship between $R$ and $\Delta T$ for a spherical tip shape using a method similar to that described by Burden and Hunt [90]. As a result, Hunt was able to obtain a relationship between $\Delta T$ and $\lambda_1$. By assuming the dendrite tip advanced at the minimum undercooling (extremum condition—Figure 2), Hunt obtained a relationship between $\lambda_1$ and the process variables for the low and high $V$ regimes as follows:

low $V$ regime,

$$\lambda_1 = 2.83\left(\Delta T_0 D \Gamma k - \frac{kGD}{V}\right)^{0.25} V^{-0.25} G^{-0.5}, \tag{9}$$

high $V$ regime,

$$\lambda_1 = 2.83(\Delta T_0 D \Gamma k)^{0.25} V^{-0.25} G^{-0.5}, \tag{10}$$

where, $\Gamma$ is the Gibbs–Thomson coefficient; $\Delta T_0$ is the equilibrium solidification range $(-m(1-k)C_0/k)$; $m$ is the slope of the liquidus.

In 1981, Kurz and Fisher (K–F) [46] proposed a new general framework for relating $R$, $\Delta T$, and $\lambda_1$ in alloy dendritic growth. The researchers used a simplified version of the Ivantsov solution (Equation (3)) to determine the solute-diffusion-limited growth of the dendrite tip i.e., the steady state interface shape. They applied the marginally stable tip operating point (Equation (7)) and developed a relationship between $R$ and the process variables. The researchers approximated the dendritic interface shape as an ellipsoid and the local arrangement as hexagonal. In this way, the $R$ could be related to the minor and major axes of the ellipsoid and to the $\lambda_1$ through geometric considerations. K–F calculated $\lambda_1$ for the low and high $V$ regimes as follows:

low $V$ regime,

$$\lambda_1 = \left[\frac{6\Delta T'}{G(1-k)}\left(\frac{D}{V} - \frac{\Delta T_0 k}{G}\right)\right]^{0.5}, \tag{11}$$

high $V$ regime,

$$\lambda_1 = 4.3\Delta T'^{0.5}\left(\frac{D\Gamma}{\Delta T_0 k}\right)^{0.25} V^{-0.25} G^{-0.5}, \tag{12}$$

where, $\Delta T'$ is the difference between the non-equilibrium solidus and the tip temperature $(T_L - T_e - \Delta T)$; $T_L$ is the liquidus temperature; $T_e$ is the eutectic temperature. K–F compared their model to Trivedi's results [91] for isolated dendrite growth operating at marginal stability and found major differences occurring at high $V$ due to the simplified approximation of the Ivantsov equation. However, even with major simplifications, they were able to match $\Delta T$ and $R$ predicted by Trivedi's model with reasonable accuracy. The application of the marginal stability criterion within the K–F model avoided predicting dendrite tip radii too small, a problem usually associated with the extremum condition (Figure 2). K–F also predicted a new phenomenon, that the $\lambda_1 - V$ relationship was split into two regimes, one for cells at low $V$ and one for dendrites at high $V$.

### 3.2. Precisely Controlled Experiments

In 1982, Mason et al. [92] performed some very careful constrained growth experiments on Pb–Sn alloys. These researchers explained how previous directional solidification experimental analysis was carried out over a limited range $V$, $G$, and $C_0$. Therefore, no consistent data existed which enabled accurate prediction of $\lambda_1$ as a function of these process variables [39]. They concluded that

the customary relationship found in the literature, $\lambda_1 = KG^{-a}V^{-b}$, implied the effects of $C_0$, $G$ and $V$ on $\lambda_1$ were independent of each other. However, their results suggested the effects of $G$ and $V$ were strongly coupled.

Following this, K. Somboonsuk et al. [93] performed further constrained growth experiments, this time using the most completely characterised material used in dendritic growth experiments, succinonitrile (SCN). The purpose was to carefully characterise the relationship between $R$ and $\lambda_1$ as functions of $V$ and temperature in the liquid. Their results concluded that when the solid–liquid interface is an isoconcentrate, the dendrite tip maintains a parabolic shape. Three important conclusions from their work were reached: (1) experimentally obtained $R$ must match those predicted by an isolated dendrite model; (2) a sharp decrease in $\lambda_1$ occurs at the dendrite-to-cell transition $V$, which corresponds to a minimum in solute Péclet number; (3) in the high $V$ regime, a variation in $G$ significantly alters the $\lambda_1$ but has no effect on $R$ or secondary dendrite spacing, $\lambda_2$; their experimental results are illustrated in Figure 5.

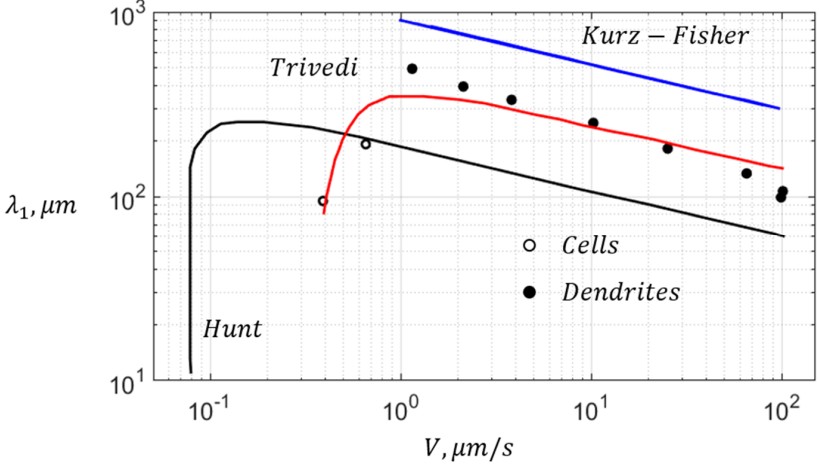

**Figure 5.** Comparison between Hunt [45], Kurz–Fisher [46] and Trivedi [47] primary spacing, $\lambda_1$, models using experimental data from Somboonsuk et al. [93]. Plot adapted from [47].

### 3.3. An Improved Analytical Primary Spacing Model

In 1984, Trivedi [47] compared the models of Hunt and Kurz–Fisher with experimental data of K. Somboonsuk et al., [93] (Figure 5). At high $V$, both models demonstrated the same functional relationship of the variables $V$ and $G$, and the Kurz–Fisher model predicted the dendrite-to-cell transition accurately. However, the experimental line had a higher gradient than that predicted by either theory. Furthermore, the Kurz–Fisher model predicted a primary spacing 1.3–2 times too high and the Hunt model 1.8–3 times too small. In addition, the relationship $V^{-\frac{1}{4}}G^{-\frac{1}{2}}$ was shown to not characterise the $\lambda_1$ in the low $V$ regime where cells are formed, hence, a more complicated equation in the form of $V^{-x}G^{-y}$ was required.

To provide a better description of the experimental results of Somboonsuk et al. [93], Trivedi [47] modified the $\lambda_1$ and $R$ relationship derived by Hunt [45] with the marginal stability criterion (Equation (7)). Following this, Trivedi determined $R$ using a quadratic equation developed earlier for isolated dendritic growth [91] that related $P_c$ to $V$, $G$ and $C_0$. For a given value of $P_c$ the author obtained two stable combinations of $V$ and $R$, whereby one set corresponded to the dendritic region and the other to the cellular. Trivedi's steady state primary spacing model is as follows:

$$\lambda_1 = 2.83(\Delta T_0 k D \Gamma L)^{0.25} V^{-0.25} G^{-0.5}, \tag{13}$$

where, $L = \frac{1}{2}(l+1)(l+2)$ for the spherical approximation of the dendrite front (Figure 3b); $l$ is the harmonic of perturbation which equals six for the dendrite growth process [66,93]. Trivedi's model

demonstrated good agreement with the results of K. Somboonsuk et al. [93] (Figure 5). The model predicted the maximum $\lambda_1$ as a function of $V$ with good accuracy. Furthermore, it demonstrated that the dendrite-to-cell transition occurred when the thermal gradient effect became significant i.e., when the Péclet number increased towards infinity, which was accompanied by a reduction in $\lambda_1$. However, Trivedi's model deviated from experiment at high $V$ and lacked a proper theoretical treatment of the cell-to-dendrite transition. Trivedi [47] summarised two important aspects for further theoretical attention: (1) a more accurate relationship between $\lambda_1$ and $R$ was required to properly predict the slope of the log $\lambda_1$ versus log $V$; (2) a theoretical description of the dendrite and cell characteristics in the region where both of these morphologies exist.

### 3.4. The Stable Range of Primary Spacing

In the past, many authors concentrated on measuring average values of $\lambda_1$ for a fixed growth condition and comparing with a unique theoretical value [94,95]. However, a growing body of theoretical and experimental evidence regarding cellular spacings indicated that a wide range of stable spacings were possible for a given set of growth conditions [93,96–99]. In fact, wide distributions of $\lambda_1$ were regularly observed across transverse sections in metallic alloys [100,101].

In 1990, Warren and Langer (W–L) [102] proposed there existed a continuous range of physically allowable $\lambda_1$, therefore, removing the necessity for an additional $\lambda_1$ selection condition to distinguish one unique $\lambda_1$. They suggested the final $\lambda_1$ depended upon the sequence of events by which the system is set into motion. The researchers set up a periodic array of dendrites consistent with both microscopic solvability theory and the macroscopic equations for the solute diffusion field. W–L then applied a Mullins and Sekerka [70,71] type linear stability analysis to a planar solid–liquid interface and calculated the crossover wavelength for the onset of instability i.e., the lower $\lambda_1$ stability bound. Their model demonstrated a reasonable agreement with the experimental results of Somboonsuk et al. [93]. However, it was only applicable when weak local dendritic coupling occurred i.e., at moderate growth velocities. In 1993 [103], W–L modified their earlier model to follow the formation of a dendritic array from the initial instability of a planar solidification front to the selection of a final steady state array $\lambda_1$ (Figure 6). They examined the morphological stability of the leading edge and used nonlinear terms to describe the development of the dendritic array. Three sequential mechanisms were incorporated into their work: (1) buildup of a solutal boundary layer in front of an initial flat interface; (2) onset of morphological instability and formation of a relatively fine array of dendritic tips; (3) array coarsening. The modified model predicted the moderate $V$ lower stability bound for the experimental results of Ding et al. [104] with excellent accuracy. However, it lacked a theoretical treatment of cellular formation or inclusion of the mechanisms behind tertiary arm branching (Figure 9b). Although it had many approximations and theoretical assumptions, their work determined some of the key aspects of non-linear interface stability analysis and predicted a new phenomenon; spatial period-doubling bifurcation (the W–L overgrowth condition).

*(a)*            *(b)*

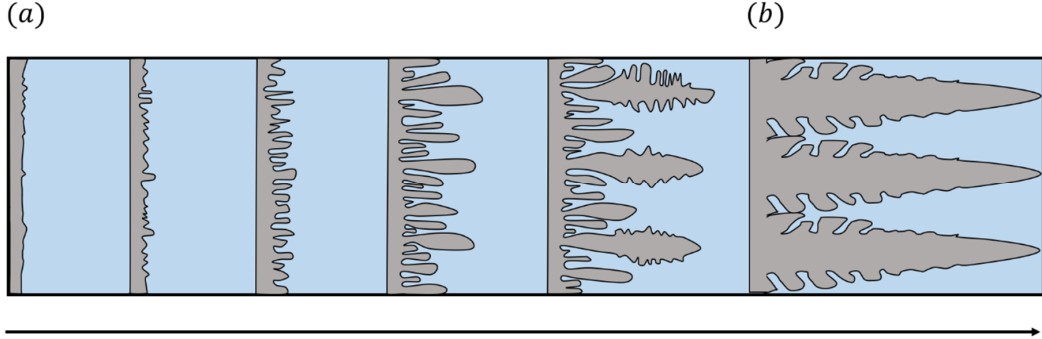

*Time*

**Figure 6.** Schematic of the development of an (**a**) initial instability of critical size into a (**b**) final steady state dendritic pattern with uniform array $\lambda_1$. Schematic adapted from [103].

### 3.5. The First Numerical Primary Spacing Model

In 1992, based on earlier developmental work [105,106], Lu and Hunt (L–H) created the first truly numerical primary spacing model [107]. Originally the model was developed to study cellular arrays, however, it was extended to dendrites when it appeared to be making correct predictions. The model used a simple, fully implicit, control volume method [108,109] to solve the diffusion problem and maintain a self-consistent dendrite interface shape. By including a non-zero surface energy they found a singular solution to the dendritic growth problem; confirming the importance of anisotropy on selection of the tip radius as developed separately by the microscopic solvability approach (Equation (8)). L–H provided the first theoretical framework to calculate the lower primary spacing bound for a fixed set of growth conditions by considering transport of solute between multi-cell walls (Figure 7b) and determined the upper spacing bound by multiplying minimum array spacing by a factor of two (Figure 8). The idea was that a local $\lambda_1$ that is twice the lower stability limit would reduce solute interaction between neighbours sufficiently to allow a tertiary arm to catch up with the growth front (Figure 9b). No model beforehand [45–47] had been able to predict the distribution of $\lambda_1$ without making arbitrary assumptions about the selection procedure or without using an oversimplified solution to the diffusion problem [107]. Their model successfully predicted the onset of the constitutional and absolute stability limits, the formation of cells at low and very high $V$, and the cell–dendrite–cell transition velocities. Furthermore, the model predicted that a range of stable $\lambda_1$ existed for cells and dendrites around the transition velocity.

(a) *Unstable Growth*                    (b) *Stable Growth*

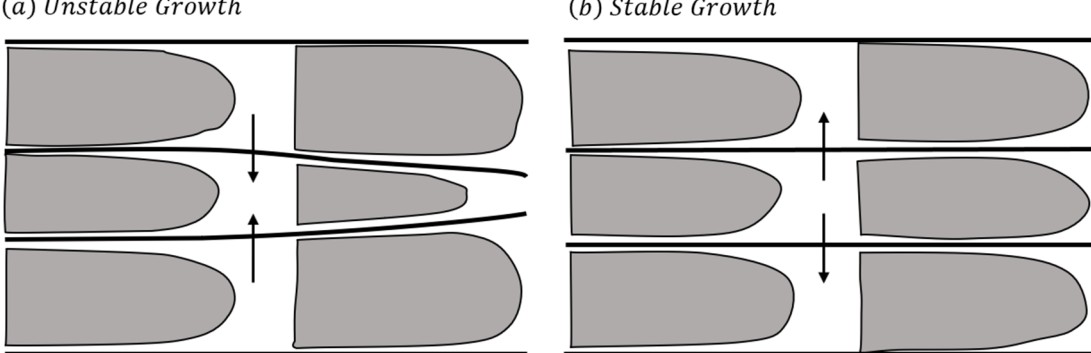

**Figure 7.** Schematic illustration of the direction of solute flow at the cell walls for (**a**) overgrowth and (**b**) array stability. Schematic adapted from [107].

In 1994, L–H extended their earlier numerical model [107] into the high $V$ region by considering a variable $k$ and non-constant liquidus slope [110]. Additionally, these researchers included a modified undercooling equation where $k$ was adjusted for solute trapping at high $V$. The $R$ and $\Delta T$ values from the numerical results were compared to the analytical KGT dendritic tip model for rapid solidification [111] and demonstrated a reasonable agreement. Following this, by suitable nondimensionalisation, Hunt and Lu [49] gave analytic expressions to fit their numerical results [107,110].

For cellular growth,

$$\lambda_1' = 4.09k^{-0.485}V'^{-0.29}(V'-G')^{-0.3}\Delta T_s'^{-0.3}(1-V')^{-1.4}, \tag{14}$$

for dendrite growth,

$$\lambda_1' = 0.7798\times10^{-1}V'^{(a-0.75)}(V'-G')^{0.75}G'^{-0.6028}, \tag{15}$$

where,

$$\lambda' = \frac{\lambda\Delta T_0}{\Gamma k}, \qquad G' = \frac{G\Gamma k}{(\Delta T_0)^2}, \qquad V' = \frac{V\Gamma k}{D\Delta T_0}, \qquad \Delta T_s' = \frac{G'}{V'} + V'^{0.333},$$

and

$$a = -1.131 - 0.1555 \log_{10}(G') - 0.007589 \big[ \log_{10}(G') \big]^2.$$

In Equations (14) and (15), the dimensionless primary spacing, $\lambda'$, refers to the radius rather than the diameter. Following this, Hunt and Thomas [112] modified Equation (15) for the case of alloy systems where $k \to 0$ as it was found that the reformulated expression behaved more realistically. According to Hunt [113], the minimum primary spacing or array stability limit is now given by the smaller of the following two equations:

$$\lambda'_1 = 2.5V'^{-b} \left( 1 - \frac{G'}{V'} \right)^{0.5} G'^{-\frac{2(1-b)}{3}}, \tag{16}$$

or,

$$\lambda'_1 = 12V'^{-1}, \tag{17}$$

where,

$$b = 0.3 + 1.9G'^{0.18}.$$

The analytical results provided an insight into the directional dendritic growth processes–property relationships and enabled theory to be compared quickly with experimental results. The Hunt–Lu model can predict $\lambda_1$ under a range of growth conditions for a variety of alloys (Figure 8). However, as this model was originally developed to study cellular growth (Figure 7), it lacks the physical mechanism to determine the upper spacing limit by tertiary arm branching (Figure 9c).

### 3.6. Understanding the Primary Spacing Selection Process

Following the work of Warren and Langer [103] and Lu and Hunt [107], Huang et al. [114] set out to experimentally determine the lower and upper $\lambda_1$ stability limits for a wide range of $G$ and $V$. To find the upper $\lambda_1$ bound for a fixed set of conditions, the researchers first formed a stable dendritic array (Figure 9b) and then step-increased the pulling velocity, $V_p$, until the array became unstable; ensuring enough time at each velocity step for array stability. For the lower $\lambda_1$ stability limit, Huang et al. applied the same method, however this time using a step-decreasing $V_p$. Their findings suggested that the lower and upper $\lambda_1$ stability limit for a fixed set of growth conditions are absolute, but the average array $\lambda_1$ is remarkably history-dependent. Furthermore, they found good agreement between their experimentally determined lower $\lambda_1$ stability limit and that predicted by the modified W–L model [103]. The history-dependent results of Ding et al., [104] are plotted in Figure 8 against the most sophisticated $\lambda_1$ models to illustrate current predictive capability.

In 1994, Han and Trivedi (H–T) [48] performed constrained dynamic growth experiments on a SCN-acetone system to understand the $\lambda_1$ selection process. The researchers confirmed the results of Huang et al. [114,116], that a range of stable array $\lambda_1$ can exist for a given set of experimental conditions. Below the lower stability limit the array is unstable and dendrite elimination occurs by overgrowth increasing the local spacing (Figure 9a). Above the upper stability limit, tip-splitting for cells or new dendrite formation by tertiary branching occurs decreasing the primary spacing (Figure 9c).

These researchers observed a new time-dependent array adjustment mechanism, which refined the Gaussian distribution of array $\lambda_1$ with time by slow lateral dendrite migration and defect interaction with the growth front (Figure 10). H–T suggested that due to the sluggish nature of this readjustment mechanism, it may not be possible to reach a steady state uniform spacing (Figure 6b) within the finite time of a given experiment. To understand the factors that may be important in determining stable $\lambda_1$ range for a given set of growth conditions, a simple global mass balance model was developed. They assumed some arbitrary interface shapes and developed a general relationship between $\lambda_1$ and the characteristics lengths:

$$\lambda_1 = A[l_D]^{0.25}[l_T]^{0.5}[d_0]^{0.25}, \tag{18}$$

where,

$$A = \frac{\alpha}{\sqrt{k}}\left[\frac{1}{2\sigma^*}\right]^{0.25}, \ l_D = \frac{2D}{V}, \ l_T = \frac{mC_t(1-k)}{G}, \ d_0 = \frac{\Gamma}{mC_t(1-k)},$$

where, $A$ is a proportionality constant; $l_D$ is the solutal length; $l_T$ is the thermal length; $d_0$ is the capillary length. Equation (18) gave clear insight into the role of the experimental variables on $\lambda_1$ selection at low velocities. The equation indicates that a change in $G$ only influences $l_T$, whereas a change in $V$ alters only $l_D$, and that both $d_0$ and $l_T$ are influenced by a change in composition as a result of a change in the solidification range. Furthermore, the theoretical analysis of H–T demonstrated that the primary spacing was proportional to the geometric mean of $R \times l_T$.

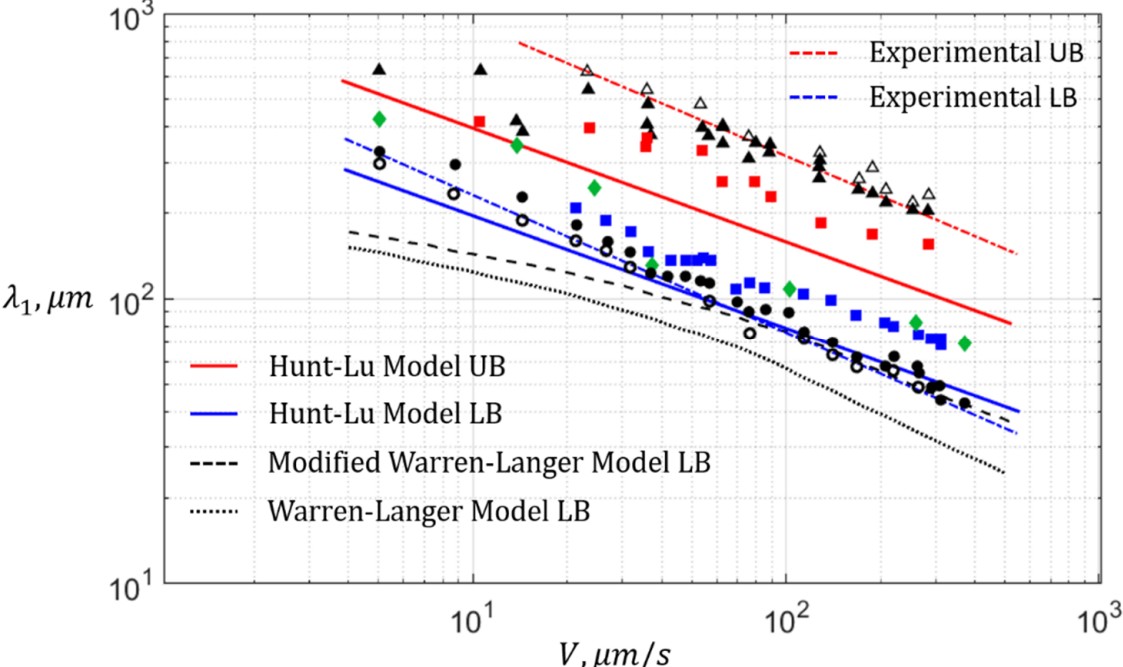

**Figure 8.** The allowable distribution of primary spacing, $\lambda_1$, within a succinonitrile-1.0 wt% acetone dendritic array, thermal gradient = 4.0 $K/mm$ [115]. ($\Delta$) critical $\lambda_1$ before tertiary arm branching; (▲) maximum stable $\lambda_1$; (○) critical $\lambda_1$ before overgrowth; (●) minimum stable $\lambda_1$; (■), (◆) and (■)—average $\lambda_1$ from step-increasing, constant and step-decreasing pulling velocity, respectively; UB—upper $\lambda_1$ bound; LB—Lower $\lambda_1$ bound. Plot adapted from [104].

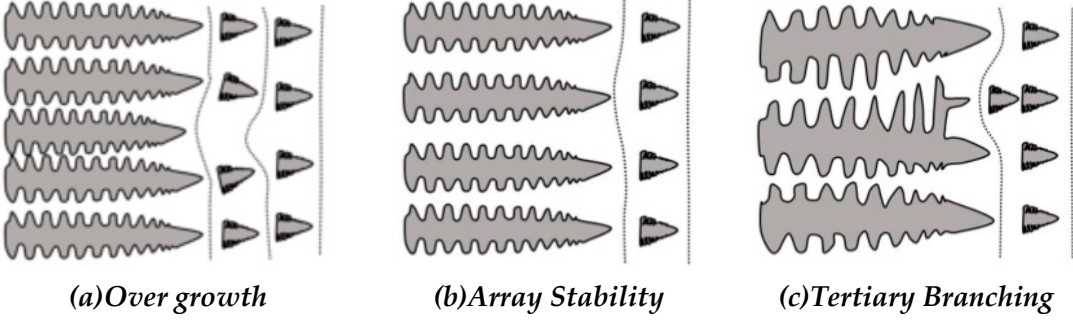

*(a)Over growth* 　　　　　　 *(b)Array Stability* 　　　　　　 *(c)Tertiary Branching*

**Figure 9.** Schematic illustration of spacing adjustment mechanism for dendrites. Dotted line indicates the liquidus isotherm. Schematic adapted from [107].

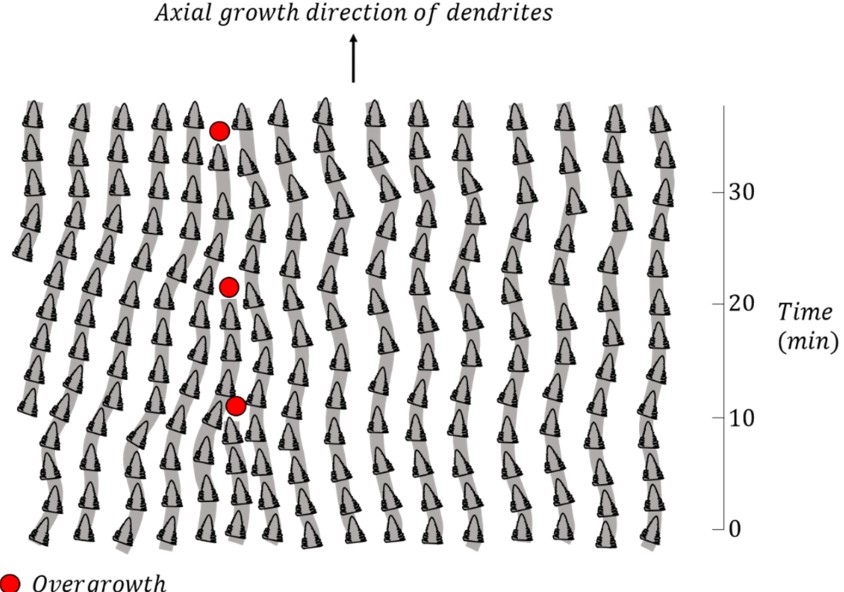

*Axial growth direction of dendrites*

● *Overgrowth*

**Figure 10.** A schematic of real dendritic tip positions in an array as a function of time. Schematic adapted from [48].

### 3.7. Delimiting the Wide Range of Stable Primary Spacings

In 1996, Gandin et al. [41] noticed the upper $\lambda_1$ stability limit still required a proper theoretical treatment in order to delimit the wide range of possible spacings [115,117,118]; clearly illustrated in Figure 8. They recognised that most prior $\lambda_1$ models ignored the effect of local misorientation, side branch formation, and tertiary arm growth between neighbouring dendrites. This is important, as tertiary arm branching determines the upper stable $\lambda_1$ before the growth of a new primary dendrite (Figure 11). Gandin et al. developed a simple analytical model based on dynamic side branch formation using the parabolic dendrite growth model [39,72,111] to compute the $V - R$ and $V - \Delta T$ relations of secondary and tertiary branches. They applied secondary arm scaling laws as proposed by Esaka et al. [119] and Somboonsuk et al. [93], and used an experimentally determined exponent by Huang and Glicksman [66] for the active branches below the dendrite tip. Their model was as follows:

$$\lambda_1 \propto \Delta T_0^a V^{-b} G^{-c} F(\theta), \tag{19}$$

where,

$$F(\theta) = 1 + d\left[(cos\theta)^{-e} - 1\right], \tag{20}$$

and $a = b = 0.25$, $c = 0.5$, $d = 0.15$, $e = 8$ and $\theta$ is the angle between the misorientated primary dendrite trunk and the thermal direction (Figure 11). The analytical branching-based model illustrates the trends of primary spacing evolution with orientation for different imposed values of $G$ and $V$. It was the first quantitative approach to evaluate the $\lambda_1$ based on the tertiary arm mechanism and emphasised the concepts required in developing a branching limited primary spacing evolution theory. However, their approach was limited to applications at locally misorientated boundaries (Figure 11) and provides only a measure of the influence of local misorientation on upper $\lambda_1$ selection.

Losert et al. [117,120,121] investigated quantitatively the linear stability analysis of Warren and Langer [102,103] and their prediction of spatial period-doubling bifurcation. Using SCN doped with 0.43 wt% of the laser dye coumarin 152, these researchers set up a steady state uniformly spaced array and slowly reduced the $V_p$ until the array became unstable (Figure 12). They observed, in the central region of the sample, every other dendrite falling back, and a doubling of the interdendritic spacing (Figure 12b). Immediately after the doubling, there was a large distribution of $\lambda_1$ which then proceeded to decrease by the lateral adjustment mechanism [48] (Figure 10). Losert et al. concluded

that a steady state dendritic array of given $\lambda_1$ (Figure 12a) is stable over a wide range of $V_p$ with a lower stability bound determined by the spatial period-doubling instability (Figure 12b). Furthermore, the same instability sets the lower bound for an array of different $\lambda_1$ for a fixed $V_p$.

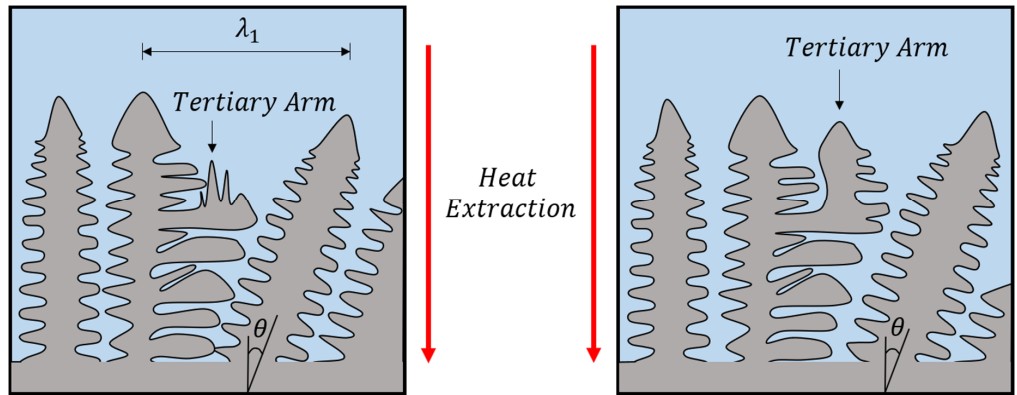

**Figure 11.** Tertiary side branch formation leading to a new primary dendrite at a divergent grain boundary. The larger the local misorientation angle, $\theta$, the larger the local $\lambda_1$. Schematic adapted from [41].

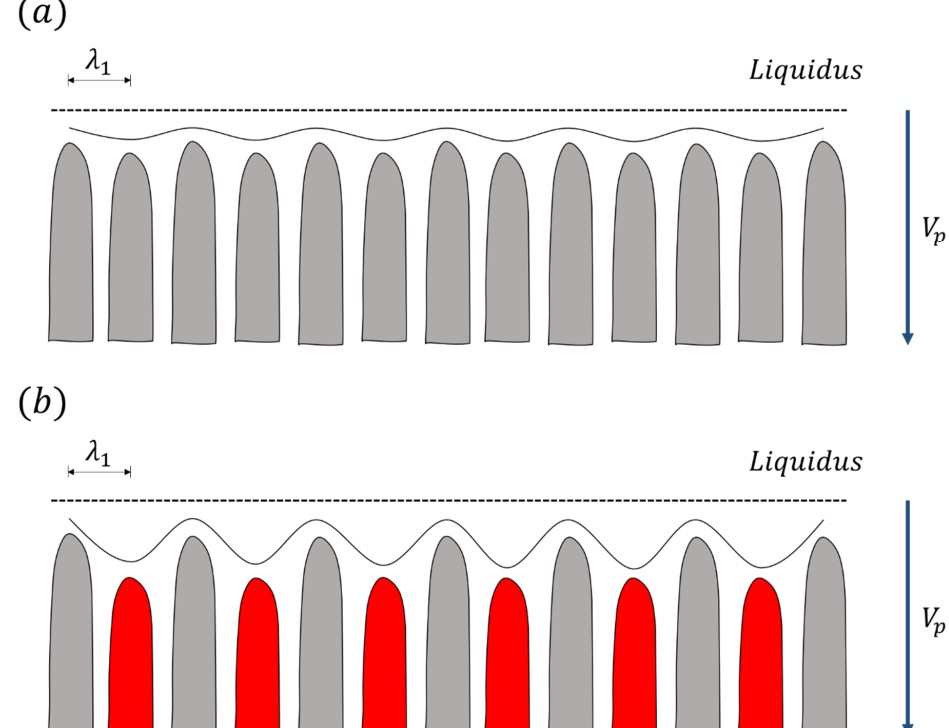

**Figure 12.** (**a**) An array of fixed $\lambda_1$ stable for a range of pulling velocities, $V_p$. (**b**) At some critical $V_p$, the spatial period-doubling instability occurs, which results in the removal of every other cell/dendrite (red) and a doubling of $\lambda_1$. Schematic adapted from [117].

### 3.8. Models Viewed as Complimentary to Hunt and Lu

In 1997, Bouchard and Kirkaldy (B–K) [50,122] developed a numerical model to characterise $\lambda_1$ for unsteady (ingot solidification) and steady state (directional solidification) heat flow conditions. These researchers were interested in unsteady state heat flow conditions as this type of solidification encompasses most industrial processes. Their model is derived for two-dimensional cells from an earlier semi-empirical heuristically developed steady state formula [123], which after modification, demonstrated its utility in the unsteady regime:

$$\lambda_1 = \alpha_1 \left( \frac{16C_0^{0.5} G_0 \varepsilon \Gamma D}{(1-k)mGV} \right)^{0.5},$$ (21)

where, $G_0 \varepsilon$ is a characteristic parameter $(600 \times 6 \; K/cm)$; $\alpha_1$ is the primary spacing calibrating factor. The model demonstrated a reasonable fit with a variety of binary alloy data but required $\alpha_1$ to account for some uncertainties. B–K suggested that Equation (21) could be used to predict $\lambda_1$ within binary alloys when experimental data was lacking, however, a prior knowledge of $G$ and $V$ at the dendrite tip is required.

In 1997, Spencer and Huppert (S–H) [124,125] developed a branchless $\lambda_1$ model to predict morphology characteristics of dendrite growth, such as $R$, $\lambda_1$, $\Delta T$ and overall dendrite shape. These researchers analysed different morphological and experimental length scales from the results of Somboonsuk et al. [93]. From this they noticed a natural separation and identified a characteristic small parameter to describe dendritic growth. S–H solved numerically asymptotic equations to derive an integral equation for the shape of a dendrite by considering nonlinear interactions between neighbouring dendrites and then solving for the nontrivial details of the shape. Iconoclastically, they were able to obtain a unique solution for the shape without considering sidebranches or a selection criterion based on surface energy. For a given set of experimental conditions, their integral equation had a family of solutions parameterised only by the $\lambda_1$ of the array. S–H determined the lower $\lambda_1$ by the Warren and Langer overgrowth condition [102] and the upper $\lambda_1$ by marginal stability [68,69]. Their model could not determine which $\lambda_1$ is observed in practice as the actual $\lambda_1$ is intrinsically time and history dependent [48,103,114]. The model was offered as an alternative view to the traditionally accepted theory of microscopic solvability and was demonstrated relevant at moderate growth $V$, where surface energy effects are negligible.

In 1998, Ma and Sahm (M–S) [40] developed a simple analytical model to provide a description of the variation in $\lambda_1$ as a function of $V$. M-S separated the dendritic envelope into a centre core and its sidearms (Figure 13), on the basis that the inclusion of sidearms are absent from the Hunt–Lu analysis [49]. M–S applied a simple relationship between the tip diameter, $(\varnothing = 2 \times R)$, and the side arm length, $S$ to determine $\lambda_1$ (Figure 13). To calculate $R$ for a given growth condition M–S used the marginal stability criterion [68,69]. To determine $S$ they calculated a 'free growth coefficient', $g_s$, which characterised the proportion of the free growth of the side arms compared with the entire solidification time. They determined $g_s$ as a function of system properties and processing parameters based on theoretical reasoning and careful study of experimental variables. The researchers derived equations for:

cellular growth $(S = 0)$,

$$\lambda_1 = 4\pi \left( \frac{D\Gamma}{k\Delta T_0} \right)^{0.5} \left( 1 - \frac{V_c}{V} \right)^{-0.5} V^{-0.5},$$ (22)

and dendritic growth $(S > 0)$,

$$\lambda_1 = 2\pi (kD\Gamma\Delta T_0)^{0.25} \left( 1 - \frac{V_c}{V} \right)^{0.75} G^{-0.5} V^{-0.25}.$$ (23)

M–S compared their model with the analytical expression of Hunt and Lu [49] over several alloy systems. When compared with cellular growth experimental data, both models demonstrated a reasonable fit. However, when applied to the moderate $V$ dendritic growth regime, the Ma and Sahm model appeared to be making better predictions than the more rigorously derived steady state theory of Hunt and Lu.

In 2002, Ma [126] developed an analytical model to describe the typical features of primary spacing selection, such as delayed response of $\lambda_1$ variation, the wide range in $\lambda_1$ distribution, and $\lambda_1$ history-dependence. The motivation behind this work was the absence of a model of tertiary arm

branching in a dendritic array of the same orientation. The model was developed for constrained growth using scaling laws [93,97] and marginal stability [68]. Ma determined a nominal $\lambda_1$ that provided the steady state baseline for the average array $\lambda_1$ and developed kinetic factors for the overgrowth and branching limits. The variation in array $\lambda_1$ was described by a beta distribution, which provided a sharp lower and upper $\lambda_1$ limit. Following the same procedure as Huang et al. [114], Ma step-increased the $V_p$ within the model. When the local $\lambda_1$ was larger than a critical branching limit, new primary dendrites formed and the average array $\lambda_1$ updated. The model was then applied to a step-decreasing $V_p$, dendrites whose $\lambda_1$ were below the critical overgrowth limit were eliminated. The model showed an excellent fit with the history-dependant results of Huang et al. [114] (Figure 14). It could quantitatively calculate the minimum, maximum and average $\lambda_1$ in response to a change in $V$ from an initial stage of stable growth. Following this, Ma [127] extended the model to predict the $\lambda_1$ distribution with varying $G$, whilst the $V$ remained constant. The results between both analytical models demonstrated the same intrinsic $\lambda_1$ relationship and a good fit with experimental data [114].

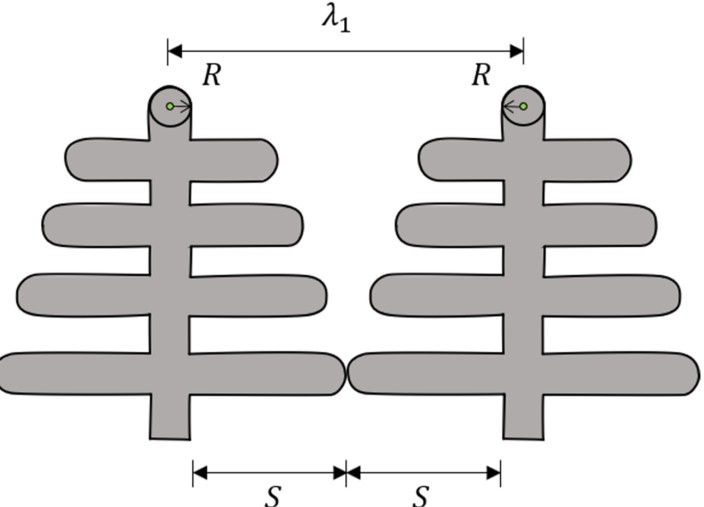

**Figure 13.** The $\lambda_1$ is the sum of the side lengths plus the dendrite core diameter ($\varnothing = 2 \times R$).

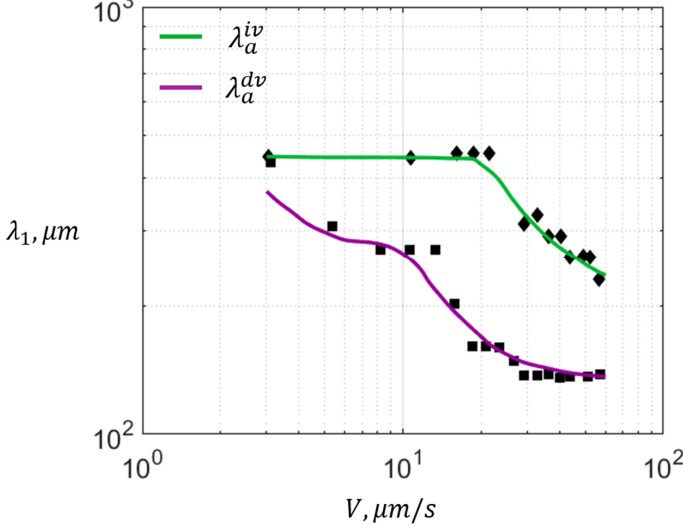

**Figure 14.** Average $\lambda_1$ predicted by the Ma model for increasing, $\lambda_a^{iv}$, and decreasing, $\lambda_a^{dv}$, pulling velocity. The model is compared with the average $\lambda_1$ experimental data for increasing and decreasing pulling velocity from Huang et al. [114]. Thermal gradient fixed at 4.8 $K/mm$. Plot adapted from [126].

## 4. Dendritic Growth Computational Modelling—Present Day

The unstable and complex morphological nature of the solid–liquid interface has been a considerable challenge for the implementation of suitable analytical and numerical models that can adequately describe primary spacing distribution as a function of the process variables. Currently, the lower stability bound for a fixed set of growth conditions has received a proper theoretical treatment and is predicted analytically [103] and numerically [107] with good accuracy (Figure 8). The upper stability bound is understood from an analytical point of view [41,126,127]. Unfortunately, the mechanisms behind local dendritic misorientation and its influence on the upper $\lambda_1$ selection procedure are not fully understood. Multiple investigation into the origins of misorientation have identified plastic deformation through differential thermal contraction in the mushy zone as the probable cause of dendritic bending [128–138]. However, to the authors' knowledge, no proper analytical or numerical treatment of the driving force behind dendritic bending or its influence on the upper primary selection procedure is reported in the literature. In addition to this, the mechanisms behind the history-dependent $\lambda_1$ selection and the stable $\lambda_1$ distribution for a given set of growth conditions are unknown. Consequently, accurate prediction of the severity of defect formation, the homogenisation times, and mechanical properties within a material, are not currently possible.

Now, the problem with applying purely diffusion-limited models to study directional dendritic array growth is the assumption of a 'convection free' environment where an isoconcentrate exists ahead of the growing solidification front. In a situation where convective effects are minimised, such as in thin samples, low Péclet number materials, and in microgravity, a steady state approach such as that of Hunt and Lu [107] may be appropriate. However, within constrained 3D dendritic growth, unsteady state heat flow conditions, natural convection, and macrosegregation can occur, significantly influencing the final microstructural patterns [39,139–144]. Unfortunately, the presence of density variations within the melt drives convective heat and mass transport. Due to the added length and time scales, microstructural patterns can differ greatly from those generated using purely heat and mass transfer [145]. The influence of thermosolutal convection within the melt can trigger unexpected and complicated flow phenomena. The resultant redistribution of heat and mass significantly influence the growth of the solidifying dendritic array, leading to macrosegregation within the solidified crystal and array disorder [39]. Fortunately, in the last 20 years, substantial increases in computational power have permitted the development of more complex modelling methods, enabling the full dendrite growth problem including convective effects to be studied. The two most popular techniques are the Cellular Automata (CA) and Phase Field (PF) methods. The purpose of the following section is to provide a very brief overview on the recent applications of modelling and highlight its expanding role within the field of solidification science.

CA has found application in realistic computation at the micro-macroscopic scale with moderate demand for computation (Figure 15). It offers a balance between computational efficiency and physically reasonable rules for modelling solidification of a domain. CA works by combining analytical microscopic dendritic growth models (e.g., LGK [146] or KGT [147]), with numerical models for heat and mass transport at the scale of grains. It is now fully coupled with momentum, mass and energy transportation in liquid, solid and mushy zones [148,149]. Models derived from CA approaches such as Cellular Automata Finite Difference (CAFD) have simulated the evolution of dendritic morphology during alloy solidification in the presence of melt convection [150–152], stray grain formation [153,154], multiscale modelling [155], geometry-related grain boundary formation [156], freckle formation [157] and multi-component systems [158,159]. However, a major challenge with this method is the substantial anisotropic influence of the underlying grid on the simulation results [160]. The grid anisotropy superposes the physical anisotropy and therefore impedes the interpretation of the simulated microstructure; currently restricting CA simulation to a qualitative representation of dendritic solidification.

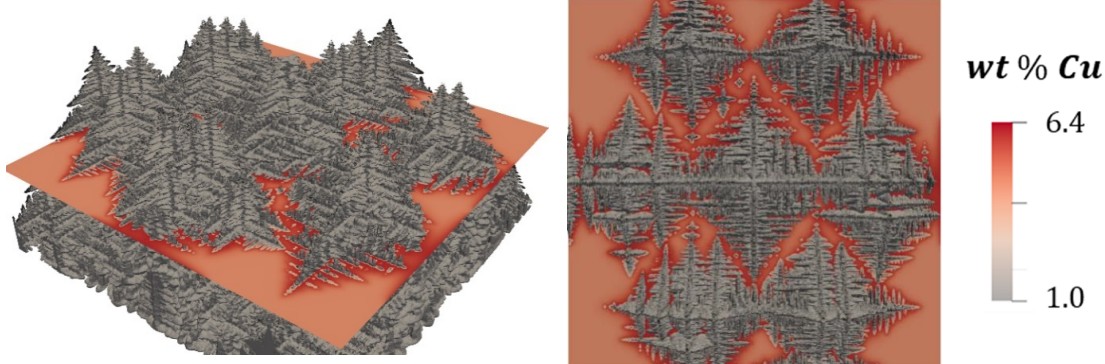

**Figure 15.** Cellular Automata Finite Difference (CAFD) model of hexagonally packed $Al - 4.5 \, wt\% \, Cu$ dendrites. Tip undercooling, $4 \, K$; thermal gradient, $1 \, K/mm$; growth velocity $0.7 \, mm/s$. The crystals were grown using the Dong et al. [161] methodology.

PF is constructed from basic thermodynamical and fundamental conservational laws and has a phenomenological character. It has developed into an important and extremely versatile technique for simulating microstructure evolution at the micro-mesoscale. The PF method possesses a significantly higher resolution of both the simulation domain as a whole and the solid–liquid interface than CA; thus, generating a substantially lower anisotropic error. PF can compute realistic and complex interface shapes associated with dendritic growth without making any a priori assumptions on the shape of the grains (Figure 16). Owing to this, PF has helped validate the microscopic solvability theory [162] and has successfully simulated a wide range of solidification and interfacial pattern formation phenomena, such as, grain growth competition [163–165] at divergent [166,167] and convergent [168,169] boundaries, spacing evolution [34,170], pattern selection [38,171–178], side branching [179], convection [145,149,180], rapid solidification [181–185] and multiphase and multicomponent systems [186–200]. In the last decade, microstructures at physically relevant length and time scales in 3D [201–204] have been simulated using advanced GPU's. However, as PF models deal with a large number of complex nonlinear terms, even with parallel architectures [202], currently only a few hundred columnar dendrites formed in diffusive conditions have been simulated [203].

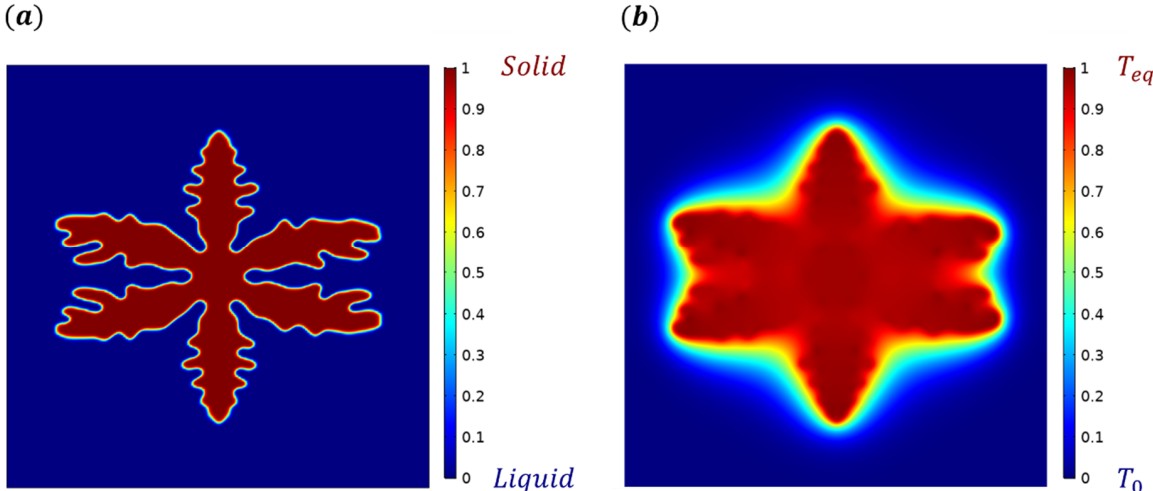

**Figure 16.** (**a**) A Phase Field model of the solidification of a hexagonal anisotropy crystal. (**b**) Dimensionless temperature variation generated from the latent heat of (**a**); $T_0$ is the characteristic cooling temperature; $T_{eq}$ is the equilibrium temperature. The crystal was grown using the Kobayashi dimensionless model with $\delta = 0.040$ and $K = 2$ [205].

## 5. Dendritic Growth in Metallic Alloy Systems—Synchrotron X-ray and Neutron Source

Accurate prediction of constrained 3D metallic array growth with analytical or numerical primary spacing models, CA or PF is difficult. These models contain mathematical simplifications, assumptions, conjectures, or phenomenological parameters that are determined from either theoretical information or analogue experimental data. Application of organic analogues in microgravity has validated the diffusion theory of dendritic growth [83–88] and elucidated the constrained 3D array growth behaviour under diffusive conditions [172,206–211]. Unfortunately, analogues cannot accurately represent the growth of a solidifying metal as they have different interfacial energies, diffusion coefficients, heat capacities, thermal conductivities, and chemical potentials. Over the last twenty years, two important experimental types have come to the forefront as fundamental tools for studying metallic systems. These techniques take advantage of the penetrating nature of synchrotron and neutron radiation for investigation into material structure and property.

Synchrotron X-ray experiments have helped validate numerical and computational modelling efforts of metal alloys [212–216]. They have provided quantitative data to test theory [217–220], however, the low penetration depth of X-rays within metals has limited synchrotron X-ray analysis to only a few hundred microns [221]; enough volume for investigation into the growth of a few metallic dendrites. It is hoped that the development of fourth-generation synchrotron light sources with multi-bend achromat synchrotron storage rings and free-electron lasers, will improve the brightness and coherent fraction of the x-ray light, whilst shortening the pulse duration [222]. However, to translate the progress in light source quality into improvements in wavelength resolution, spatio-temporal limits, and new science, requires similar progress in aspects such as X-ray optics, sample preparation, beamline technology, data analysis, and detectors [223]. Synchrotron sources have the potential to make great contributions in understanding the mechanisms behind pattern formation in constrained growth metallic alloy systems. However, to exploit them, the full technology from source to detector must first be developed to increase voxel size and enable the investigation of larger dendritic arrays.

More recently, there has been a revived interest in neutron imaging due to improvements in computer processors and bus speeds, large fast data storage devices, CCD imaging chip development and low-light cameras. Although neutron imaging cannot compete with X-rays in terms of spatial or temporal resolution [224–226], they are especially good at probing objects made from heavier elements and can penetrate thick component sections enabling investigation of bulk material characteristics. Recently, a micro channel plate (MCP) detector was combined with a Medipix2/Timepix readout which improved neutron spatial resolution from 2 *mm* to 55 *μm* [227,228] for a field of view of several square centimetres. Following this, multiple new imaging and diffraction methods have been developed to investigate the shape and orientation of grains [224,229,230]. The spatial resolution of the MCP detector is now nearly ten times smaller than the typical $\lambda_1$ length [231,232]. Therefore, neutron imaging may provide the micro-macroscopic experimental link for improving understanding of the process versus property relationships within directionally solidified bulk metallic alloys.

## 6. Outlook

This review summarises the important aspects of directional dendritic growth and primary spacing selection within the past 85 years. In this time, there have been significant developments in understanding the dendritic growth problem. Extensive theoretical and experimental studies have established criterion by which dendrite tip radius is selected under given experimental conditions. This paved the way for analytical and numerical models of primary spacing evolution and the confirmation of the diffusion part of the dendrite growth theory. More recently, the Cellular Automata (CA) and Phase Field (PF) techniques have enabled study of the influence of convection on dendrite growth evolution. Complimentary synchrotron X-ray and neutron radiation experiments have enabled quantitative testing of theory and numerical modelling on metallic alloy systems. In the authors' opinion, constrained growth research should focus on:

- providing a proper theoretical treatment of the mechanism/s behind the upper primary spacing stability bound under 3D growth conditions;
- determining the reason behind the history dependence of primary spacing distribution for a given set of growth conditions;
- investigating 3D lateral translation and providing a mechanism behind this phenomenon;
- understanding the relationship between strain and misorientation;
- quantifying the influence of convection within the melt on primary spacing variation;
- comparing microgravity experiments and PF modelling of diffusive directional dendritic growth with identical terrestrial experiments;
- reducing the influence of grid anisotropy in CA to enable quantitative modelling of micro-macroscopic scale phenomena;
- improving computational efficiency of PF modelling to facilitate larger scale simulations with convective melt flow;
- utilising synchrotron X-ray and neutron sources for further investigation into directional dendritic growth theory and ascertaining metallic phenomenological parameters.

**Author Contributions:** Conceptualisation, investigation, and writing—original draft presentation, J.S.; writing—review and editing, B.N.; supervision and funding acquisition, H.D. All authors have read and agreed to the published version of the manuscript.

**Funding:** This research and the APC was funded by EPRSC grant number EP/L016206/1.

**Acknowledgments:** J.S. and B.N. wish to acknowledge EPSRC CDT (Grant No: EP/L016206/1) in Innovative Metal Processing for providing PhD studentships for this study and Rolls-Royce Plc for providing financial support. The authors would also like to thank Jun Li and the reviewers for providing insightful comments for the manuscript revision.

**Conflicts of Interest:** The authors declare no conflict of interest.

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
