# Peer review of "On Directional Dendritic Growth and Primary Spacing—A Review"

_crystals, doi:10.3390/cryst10070627_

Round 1

Reviewer 1 Report

The manuscript attempts to present a comprehensive review of directional solidification modelling in respect to the tip growth conditions and primary arm spacing. Since Primary arm spacing is the focus this automatically implies columnar directional solidification. The review covers major contributions from the previous 85 years, starting with Papapetrou's work in 1935 to the present day work in Phase Field and Cellular Automata modeling.

The authors have selected a topic within an already crowded market of review studies. The most recent review work has been presented by Kurz, Rappaz, and Trivedi who have chronicled a two-part review in the journal International materials reviews covering all of the major work in Dendritic growth modelling from 1700 to 2000 (part I) and 2001 to 2018 (part II). Nevertheless, the existence of previous review work in a topical area should not preclude any new reviews provided that any new proposed reviews have a clear and distinct focus that may contribute to the community. 

I have read through the current review work and I have found several inconsistencies and shortcomings in this submission. I will describe these technical inconsistencies in the proceeding paragraphs.

1) Equation 1, the Ivantsov equation, only treats a 2D plate-like dendrite. This 2D model is theoretically correct, but no such 2D plate-like dendrite exists. Dendrites are 3D objects and Horvay-Cahn presented classical work that allowed for 3D dendrites to be modeled  as elliptical paraboloids and paraboloids of revolution. The authors should have at the very least presented the Ivantsov equation for a 3D paraboloid dendrite where the inamtsov solution is given as P*exp^P*E_1(P).

2) The dimensionless undercooling was given as (DeltaT*C_p)/L. This is correct for dimensionless 'thermal' undercooling only, which is only relevant for unconstrained thermal (pure) dendrites or equiaxed alloy dendrites. In the context of columnar directional solidification, there should be no thermal undercooling as constrained thermal conditions only apply. The dimensionless undercooling is solutal undercooling only and the parameter omega should be given as (c_t-c_o)/(c_t*(1-k)).

3) When describing the shortcomings of the Ivantsov solution, the authors stated: 'Eq (1) produced an infinite set of tip radii for a given growth condition' which is not that accurate. It produced an infinite set of V & R combinations that would satisfy the requirement of VR=constant but would give no unique solution of V and R for a given growth condition. The authors' wording could be improved as it is misleading as it is currently written.

4) In section 3 the authors state that: 'The variation in composition between the advancing dendrite and the surrounding interdendritic region gives rise to microsegregation in the solidified crystal, which is characterised by λ_1'. This is inaccurate, the variation in composition across the solid-liquid interface that gives rise to microsegreagation is characterised by the partition coefficient and not the primary spacing.

5) When describing the work of Hunt in calculating the primary arm spacing, the authors have incorrectly described the parameter Delta_T_o  as the undercooling at the tip, which is not the case. The parameter  Delta_T_o is given as m*c_o*(k-1)/k and is in fact the freezing range of the alloy at composition c_o, that is, Delta_T_o = T_L-T_s. 

6) When describing the work of Kurz and Fisher, the authors use Delta_T_o^(1/2)  in equations (9) an (10) where literature sources provide tip undercooling Delta_T^(1/2). This appears to be a mistake leading from a misunderstanding between the differences between freezing range and tip undercooling.

7) Equation(s) (14) which give a Buckingham PI type model (dimensionally homogeneous model) model of dendrite kinetics is not consistent with the main work in the literature (Trivedi and Kurz, Acta Metall. Mater. 42, 1994). There is no explanation provided or argument given to explain where the differences arise from. This could be seen as a major oversight of a significant literature source.

8) There are areas in the text where the authors should have paid more attention to detail in the use of language. For example, the authors describe dendrites as "tree like structures that grow parallel to the heat flow and are the predominant structures in solidification" and then go on to say "Their growth directions determined by crystallographic anisotropy, which dictates the most energetically favourable directions." There are several problems with this statement both grammatical and technical. The second sentence is incomplete as their is no clear subject-verb relationship. The term 'tree like' should be made into a compound adjective with the addition of a hyphen as follows "tree-like". But, the main problem is that the statement is contradictory or not given a clear picture of accepted understanding. Typically the dendrites do grow in the direction that is parallel but opposite to the heat flow in the <100> direction for cubic structures. However, in some cases, equiaxed dendrites have been discovered that grow in the <111> direction and the role of anisotropy strength is as yet unclear in this alternative growth direction. The growth orientation and alignment in directional solidification is more closely associated with the Walton-Chalmers competitive alignment selection criterion. 

9) Citation [1] does not seem fully relevant as a first reference because it treats the topic of columnar to Equiaxed transition in directional solidification, whereas, the issue of primary arm spacing is only relevant to columnar  solidification and not to equiaxed solidification (where average grain size is more important).

10) It is unclear what is meant when the authors state greater focus should be given to 'providing a proper theoretical treatment of the mechanism behind the upper primary spacing stability bound in unconstrained three-dimensional growth conditions'. Surely, if the growth conditions are unconstrained then equiaxed solidification would occur instead of columnar solidification - then primary spacing becomes irrelevant and would be replaced by average grain sizes in the equiaxed grain structure?

In summary and in light of the very comprehensive reviews in this area, the authors have provided a review with too many inconsistencies/shortcomings  in the explanations provided for accepted literature. The authors would be well advised to not try and compete with the highly comprehensive reviews in this area but focus on areas that have been under represented in the review literature. For example, a significant topic that would be worthy of highlighting in a review is the under-reported incongruity between predicted primary arm spacing and the measurement methods for characterizing average primary arm spacing, e.g., crystal counting, Voroni, Warnken-Reed, Minimum Spanning Tree methods, etc. Or the application of digital computational methods for characterizing primary arm spacing in directional solidification (both in-situ and ex-situ).

In light of the listed shortcomings and the high bar set by other recent reviews, unfortunately, I must recommend that this manuscript be rejected.

Reviewer 2 Report

The topic of the manuscript is very interesting and important in the field of dendritic crystallization. Presented review of knowledge regarding dendritic growth and primary spacing, makes easier to understand the problem, and directs further research. The text is well organized in chronological order from 1935 to this day and grouped into paragraphs related to the theoretical models  and experimental results. The authors made a great review of the most important publications on the issue and systematized it in a transparent way, adding theirs interesting comments.

Only a few editorial corrections should be made to the text:

- line 61: .”… Ω0 is the dimensionless … ” ?

- line 342: In figure caption there are shifts of the marks in the whole text

- line 366: the comma is in the wrong place (at the beginning of the line)

- line 418: “…characteristic…” should be: “…characteristic”

and format of the References should be corrected according to the journal guidelines.

I recommend the manuscript for publication in its current form after slight editing corrections.

Reviewer 3 Report

The authors made valuable contributions by reviewing and providing a chronological narrative on the development of the directional dendritic growth and primary spacing over the past 85 years. The manuscript is recommended to be published as it is with small changes. 

  • The first remark concerns equation 4. Please check the uncertainty value. It has three significant places but the value of stability criterion has four of them.
  • Another remark concerns the size of the drawings' and graphs' descriptions, e.g. in Figure 3 the markings a and b are too large. The same remark applies to drawings: Figure 4, Figure 5, Figure 6, Figure 7, Figure 8, Figure 13, Figure 14, Figure 16.
  • In the caption below the Figure 8, the symbols of the graphs series have shifted.
  • In addition, references 70, 71, 72 and 116 are not properly formatted.

The authors state that there are no papers on defect formation and their relation with morphology of the dendritic array. In addition, they state that this can be studied in the future only by synchrotron x-ray and neutron sources. However, there are many articles on this topic. Below I present articles only from the previous year. The articles describe, among others, the variation of the crystal orientation and the dendrite array, defects creation, lateral growth of secondary dendrite arms which may be related to the shape of the crystallization front. The analysis of the dendrite arrangement in those articles was carried out using the simplest than synchrotron X-ray experiments like scanning electron microscopy, X-ray diffraction topography, and Laue diffraction. Please read this and if possible cite.

Bogdanowicz, W.; Krawczyk, J.; Paszkowski, R.; Sieniawski, J. Primary Crystal Orientation of the Thin-Walled Area of Single-Crystalline Turbine Blade Airfoils. Materials 201912, 2699.

Huo, M.; Liu, L., Yang, W.; Hu, S.; Sun, D.; Su, H.;  Fu, H.  Dendrite growth and defects formation with increasing withdrawal rates in the rejoined platforms of Ni-based single crystal superalloys. Vacuum 2019,161, 29-36.

Hallensleben, P., Scholz, F., Thome, P., Schaar, H., Steinbach, I., Eggeler, G., & Frenzel, J.  On crystal mosaicity in single crystal Ni-based superalloys. Crystals 2019 9, 149.

Reviewer 4 Report

General comments

The paper gives a review on the directional growth, with an emphasis on relation between primary spacing and solidification parameters. The paper is very clear and presents a topic of great interest, both from fundamental viewpoint and industrial applications.

Nevertheless, the reviewer suggests to Journal Editor not to publish the paper until the following issues will be addressed:

(1)  A recent review (in two parts) on the same topic has been published in “International Materials Reviews” by Kurz et al.:

-  Kurz, W., D. J. Fisher and R. Trivedi (2019). "Progress in modelling solidification microstructures in metals and alloys: dendrites and cells from 1700 to 2000." International Materials Reviews 64(6): 311-354

- Kurz, W., M. Rappaz and R. Trivedi (2020). "Progress in modelling solidification microstructures in metals and alloys. Part II: dendrites from 2001 to 2018." International Materials Reviews: 1-47.

These papers were referred as references [79] and [88] in the present paper. It would be very important for publication to explain what is the interest or the differences of this paper compared to the two previous papers.

(2) The authors mainly focused on primary spacing but secondary spacing is also a very important parameter, in particular for microsegregation. For a sake of completeness, it seems to me that a review on secondary spacing has to be added to the paper to encompass the topic.

(3) In the abstract, the reviewer was attracted by the sentence “State of art modelling and experimental techniques are discussed within, and their limitations highlighted”. What a great disappointment then to read the two short sections devoted to these sub-topics at the end of the paper!

Additional comments:

  1. Introduction:

Page 1, line 22: Their growth directions are determined by …..

Page 1, line 23 : “Advancing dendrites form a solute profile significantly different to the nominal solute content of the alloy”, A profile is not comparable with a unique value.

The same remark for Page 6, line 173.

Page 1, line 24 : Reference [1], and then [2,3]: It seems to the reviewer that not to mention any old and seminal Tiller et al.’s paper in a review paper is a little bit cavalier.

This comment is also valid for other citations in the paper.

  1. The Operating Point of a Dendrite Tip (1935 - 1999):

Page 3, Figure 2: In Fig.2 (adapted from [63]), one can see the curve for the Temkin model. It should be interesting to add in the text one-two sentences to explain the origin of this curve.

Page 6, line 163:  The purpose was to greatly reduce the influence of heat and mass transport on dendrite tip selection.

Please correct: The purpose was to greatly reduce the influence of convective heat and mass transport on dendrite tip selection.

Page 6, line 175:  See the second issue in General comments.

  1. Primary Spacing Selection in Directional Solidification (1979 - 2004)

Page 9, line 271: Figure 8 is mentioned before Figures 6 and 7.

Page 11, line 322 : “…. case of alloy systems where k à 0, as it was found….” The authors could mention the experimental paper:

Nguyen-Thi, H., Y. Dabo, B. Drevet, M. D. Dupouy, D. Camel, B. Billia, J. D. Hunt and A. Chilton (2005). "Directional Solidification of Al-1.5wt% Ni alloys under diffusion transport in space and fluid flow localisation on Earth." J. of Crystal Growth 281: 654-668.

  1. Dendritic Growth Computational Modelling – Present Day

Page 19, line 502-504: These sentences are directly copied from reference [143] and pasted.

Page 19, line 518: “However, a major challenge with this method is the substantial anisotropic influence of the underlying grid on the simulation.” The reviewer does not agree with this sentence. It is the case for all type of simulations.

  1. Dendritic Growth in Metallic Alloy Systems – Synchrotron X-ray and Neutron Source

Page 20, line 555 : To overcome this problem, synchrotron X-ray experiments have been utilised to help validate numerical and computational modelling efforts of metal alloys [210-212] “ the reviewer does not agree with this sentence. Usually, new topics come from experimental observations!!

It is strange not to see any Mathiesen’s paper in the list of references, for instance:

Mathiesen, R. H., L. Arnberg, K. Ramsokar, T. Weitkamp, C. Rau and A. Snigirev (1999). "Time-resolved X-Ray Imaging of Dendritic Growth in Binary Alloys." Phys. Rev. Lett. 83: 5062-5065.

or a review paper on this X-ray imaging technique:

Mathiesen, R. H., L. Arnberg, H. Nguyen-Thi and B. Billia (2012). "In Situ X-Ray Video Microscopy as a Tool in Solidification Science." JOM 64(1): 76-82.

Page 20, line 561: Increasing coherent flux and extending current synchrotron with far superior wavelength resolution capabilities and large improvements on spatiotemporal limits. Where is the verb ?

The reviewer does not agree with this sentence. The main limiting device in current synchrotron facility is currently the detector, more than the X-ray beam flux.

Round 2

Reviewer 1 Report

Any review in the physical sciences literature should serve the community by being an authoritative review for the benefit for both the current and upcoming generation of researchers. As mentioned in my first review response, there are several excellent reviews in the main body of literature. Some of which are very recent and comprehensive. 

I have reviewed the updated manuscript and I still have major concerns for the quality of this review. Several aspects, if published in its present form, are unclear and will serve to confuse new researchers.

The use of language is still troubling, for example, the authors state 'At their origin, is morphological instability of the solid-liquid interface', which is a incomplete sentence. (Where is the subject-verb relationship?) But the use of the word origin is misleading because this may make an informed reader think about nucleation of the crystal. 

I am still unconvinced that the review defines constrained growth adequately when they say: 'In directional solidification, the heat flow is constrained to one direction, which results in primary dendrite tip growth aligned with the [001] plane.' This is an unsatisfactory reference to constrained growth. Tiller explained  the terms constrained and unconstrained crystal growth and it refers to the Gibbs free energy gradient (or more usefully temperature gradient) at the interface. If the crystal interface is at a lower temperature than the melt then the system is constrained, if the crystal interface is at a higher temperature than the melt then it is unconstrained. The reference to the alignment with the [001] plane is very unclear. The manuscript would benefit greatly from a clear description (with suitable graphic) describing constrained growth and its significance in directional solidification.

Reference [3] is not very helpful in this context, since it does not deal with columnar or directional solidification as a topic.

Also reference [4], in my independent viewpoint, is under-represented. Reference [4] is a highly cited experimental study on cells and dendrites, but this review overlooks the details provided in the findings from this important study. It is stated: "In the literature, these dendritic patterns are classified as either square, hexagonal, or random [4]." In ref[4], only the cellular arrays were termed hexagonal. The dendritic arrays were described as being less regular. Indeed the study showed that the B value of 0.8 for dendrites was between that for a square (B=1) and random array pattern (B=0.5). The findings from this early manuscript just highlights a point that I made in my initial review, i.e., matching predicted primary arm model outputs and the measurement methods for characterizing average primary arm spacing, e.g., area counting, Voroni, Warnken-Reed, Minimum Spanning Tree methods, etc. Ref [4] showed that even for an area counting method, the situation is not so straight forward, because the value of B required in the characterisation method changes depending on the morphology (cells or dendrites). This issue is worth point out in more detail and any review in the area needs to make this point more clearly.  

The preceding points raises a significant concern with the review. The introduction makes no clear distinction on the planar-cellular-dendritic transitions for directional solidification (and the differences between cells and dendrites, in particular). Later on in the manuscript, references are made to equations from literature for primary arms spacing in cellular and dendritic arrays under different growth regimes. But, as mentioned, the difference between a cell and a dendrite was never made clear at the introduction stage; this is is a troublesome oversight. Again, this lack of clarity will be very confusing for a new researcher to the field. I would strongly recommend that this be addressed by describing the differences between cells and dendrites. This may require a title change.

Another point, which relates to the use of language, is the overuse of the phrase  'the authors'. Usually, when this is used it refers to the point of view of authors of the current manuscript (as used, for example, in line 710). But in this manuscript the phrase is used repeatedly to refer to several different groups of authors on many different manuscripts. I would suggest reworking to avoid overuse of the term 'the authors' and reserve it for when the authors of the current manuscript give their own point of view. 

The authors has published some interesting work on automatic recognition of crystal arrays, which they cited. However, there is no mention of the recent work on skeletonisation by J. Miller et al. As an independent referee, it would be fair to include something on the skeletonistation method (cited by the authors in [234]) as it is highly relevant for characterising 3D directional solidified array structures.

For these reasons, I still have major concerns for this manuscript if it were to be published in its current form.

Author Response

In light of the most recent feedback from the reviewers, we would like to point out that there might be a possible conflict of interest with Reviewer 1. The authors would like to bring the following points to your attention:

Reviewer 1 states: "The authors would be well advised to not try and compete with the highly comprehensive reviews in this area". This is a personal opinion that is not backed up by any scientific reasoning.

The reviewer repetitively rejects the work but only highlights minor issues with language that may or not be relevant, then proceeds to write a significant piece of text to justify some apparent misunderstanding of the literature

The reviewer repetitively mentions adding primary spacing characterisation techniques into the review. Our review is focused on detailing the relationship between the primary spacing and the local/bulk solidification environment. The method by which the primary spacing is determined is not the focus of this review. The work of Tschopp et al. provides a good review for this purpose. It is worth mentioning that the method by which the primary spacing is characterised is not massively significant. Whether this is by the Voronoi, Warnken-Reed, Minimum Spanning Tree Methods or even by hand the eventual variation in measured results in small. In fact, measuring by hand is often much more accurate.

The independent reviewer has suggested the authors include the 3D characterisation work of J. Miller et al. Our review is not focused on characterisation methods but the relationship with the primary spacing and process variables. The authors find it interesting that the independent reviewer argues for characterisation methods, especially in the light that our most recent published article discusses the limitations of the skeletonisation technique described in the work of J. Miller et al.

In light of these above points the authors would ask to consider asking another new reviewer, instead of the current reviewer 1. We believe that regardless of any modifications we make to the manuscript, the reviewer 1 is unlikely to be satisfied with what ever change we will make.

Reviewer 4 Report

The reviewer greatly appreciates the authors' answers. All the issues that were mentionned in the first review have been taken into consideration by the authors and they convince me that their paper deserves to be published.

Moreover, the authors have corrected some typos and sentences in the manuscript and the manuscript has been significantly improved and now warrants publication in Crystals.

Author Response

Thanks for your comments.